# DIVERSE MACHINE TRANSLATION WITH A SINGLE MULTINOMIAL LATENT VARIABLE

## ABSTRACT

There are many ways to translate a sentence into another language. Explicit modeling of such uncertainty may enable better model fitting to the data and it may enable users to express a preference for how to translate a piece of content. Latent variable models are a natural way to represent uncertainty. Prior work investigated the use of multivariate continuous and discrete latent variables, but their interpretation and use for generating a diverse set of hypotheses have been elusive. In this work, we drastically simplify the model, using just a single multinomial latent variable. The resulting mixture of experts model can be trained efficiently via hard-EM and can generate a diverse set of hypothesis by parallel greedy decoding. We perform extensive experiments on three WMT benchmark datasets that have multiple human references, and we show that our model provides a better trade-off between quality and diversity of generations compared to all baseline methods.[1]

## 1 INTRODUCTION

Modeling uncertainty in the output distribution is a fundamental problem in machine learning. Some prediction tasks, in particular those involving text generation, may admit many valid outputs. However, in most cases the learner only observes one out of many correct outputs both at training and test time. This raises questions about how to properly fit the data distribution and how to properly measure such fitting.

In this work, we investigate these questions for the task of machine translation (MT), however the arguments and methods are general and can equally be applied to other models and tasks. Specifically, we aim at better modeling uncertainty for the purpose of effectively decoding a diverse set of hypotheses. The motivation is twofold. On one hand, we are interested in exploring the problem of modeling uncertainty, and we would like to better understand how different latent variable models capture uncertainty. MT serves as a well defined task for this investigation. On the other hand, we are driven by the potential practical applications of such models for MT, which could enable end-users to express their preference about how to translate a piece of text.

Previous work has explored approaches for heuristically modifying the decoding algorithm to produce diverse hypotheses, e.g., by enforcing diversity constraints during beam search (Li & Jurafsky, 2016; Vijayakumar et al., 2016). Nevertheless, it seems more natural to consider *explicit* ways to represent the multi-modal nature of this task by introducing latent variables in the model. Notably, Zhang et al. (2016) proposed a variational version of NMT with a latent Gaussian variable (Kingma & Welling, 2013). While they showed improvements due to the regularization effect of the Monte Carlo gradient estimate, it is unclear how well the corresponding latent space actually captures different output modes.

We propose a much simpler approach that introduces a single multinomial latent variable to capture uncertainty in the target sentence. This model, which is a simplified instance of a mixture of experts (MoE) model (Jacobs et al., 1991), is much easier to train and interpret than previous approaches. Our approach also enables fast parallel decoding from different latent values to generate a diverse set of hypotheses.

We evaluate our method on three WMT benchmark datasets that uniquely provide test sets with multiple human references. These datasets enable us to *quantitatively* evaluate diversity, and we

---

[1]Code to reproduce this work is available at: anonymized URL.

provide the first systematic comparison of the various methods that produce multiple hypotheses. Our results demonstrate that our proposed model is more robust and generates both high-quality and diverse translations, and compares favorably to variational NMT models, other variants of mixture of expert models, and diverse inference algorithms such as diverse beam search (Vijayakumar et al., 2016) and biased sampling (Graves, 2013; Fan et al., 2018). Moreover, our study provides insights for best practices to avoid several failure modes of latent variable models, in particular when these are trained with modern architectures using dropout and shared parameters.

## 2 DIVERSE DECODING IN NMT

Let $x$ be a source sentence consisting of words $(x_1, \cdots, x_L)$, and $y = (y_1, \cdots, y_T)$ a given reference (target) translation. An NMT model has an encoder-decoder structure. The encoder maps $(x_1, \cdots, x_L)$ to a sequence of representations $h = (h_1, \cdots, h_L)$ that are then fed to the decoder, which generates an output sentence one word at a time conditioned on $h$. At each time step, the decoder additionally conditions its output on the previous outputs, resulting in an auto-regressive factorization $p(y|x; \theta) = \prod_{t=1}^{T} p(y_t | y_{1:t-1}, x; \theta)$, where $\theta$ denotes the parameters of the network and is learned by maximum likelihood training.

At test time, approximate inference algorithms such as beam search are commonly used to decode the most likely translations. Beam search decodes a sequence left-to-right while maintaining the $K$ best hypotheses at each step. Although it is often effective in finding high quality hypotheses, these hypotheses typically have low diversity with only minor differences in the suffix. To remedy this, Vijayakumar et al. (2016) proposed *diverse beam search*, where they perform beam search sequentially for $G$ iterations, penalizing the selection of words used in previous iterations.

Another decoding strategy is to sample repeatedly from the model's conditional distribution $\hat{y}_t \propto p(\cdot | \hat{y}_{1:t-1}, x; \theta)$. Unfortunately, this approach usually produces diverse but low quality outputs. Several heuristic methods have been proposed to obtain better quality, such as sampling with a temperature, biased sampling from the top $k$ most likely words instead of all words at each step (Graves, 2013; Fan et al., 2018), oversampling $K' > K$ outputs and retaining the top $K$, adding noise to the hidden state (Cho, 2016), etc.

## 3 LATENT VARIABLE NMT

Rather than search for diverse outputs at test time, which may be slow and difficult to tune, we aim to explicitly model uncertainty during training. We introduce a multinomial latent variable $z$ taking values in $\{1, \cdots, K\}$ to capture various ways to translate the source sentence, and train a latent variable model $p(y|x; \theta) = \sum_{z=1}^{K} p(z|x; \theta) p(y|z, x; \theta)$. Intuitively, we want to push the uncertainty from $p(y|x)$ into $z$, so that we can easily explore multiple modes during training and inference.

For each value of $z$, let us consider $p(y|z, x; \theta)$ to be an *expert* and $p(z|x; \theta)$ to be a gating function over experts, i.e., a source-dependent prior of using expert $z$. Training proceeds by minimizing the negative log-likelihood loss function:

$$\mathcal{L}_{\text{Soft-MoE}}(\theta) = \mathbb{E}_{(x,y) \sim \text{data}} \left[ -\log \sum_{z=1}^{K} p(z|x; \theta) p(y|z, x; \theta) \right] \tag{1}$$

This is the Soft-Mixture of Experts (Soft-MoE) model. In our work, the encoder and experts are based on the Transformer architecture (Vaswani et al., 2017), while the gating is implemented via mean-pooling over the encoder's representations followed by a single layer feed-forward net to predict $p(z|x; \theta)$, similar to related work (Zhang et al., 2016).

### 3.1 OUR APPROACH: HARD-MOE

A well-known challenge in training Soft-MoE models is that experts "die" during training; in extreme cases the model collapses and only a single expert is trained (Eigen et al., 2014; Shazeer et al., 2017b). We address this by introducing two simplifying assumptions. First, we assume that the prior $p(z|x) = 1/K$ is uniform and source independent. This encourages the model to use multiple experts instead of collapsing into one with extreme $p(z|x)$ values. Second, we assume that a particular

---

**Algorithm 1** Hard-MoE Training with $K$ Experts

---

Initialize model parameters $\theta$
**repeat**
    Sample a mini-batch of $m$ examples $\{(x^{(i)}, y^{(i)})\}_{i=1}^{m}$
    Set the model in evaluation mode, turn off dropout and disable gradient computation
    **for** $z$ in $\{1, \cdots, K\}$ **do**
        Compute $-\log p(y^{(i)}|z, x^{(i)}; \theta)$
    **end for**
    Let $z^{(i)} = \arg\min_z - \log p(y^{(i)}|z, x^{(i)}; \theta)$
    Set the model in train mode, turn on dropout and enable gradient computation
    Do another forward-backward pass to compute loss $\mathcal{L} = \frac{1}{m} \sum_{i=1}^{m} - \log p(y^{(i)}|z^{(i)}, x^{(i)}; \theta)$ and
    update $\theta$ by gradient descent
**until** Convergence

---

translation $y$ is well explained by exactly one expert, that is $p(y|z, x)$ is large for only one value of $z$. This forces experts to specialize on different subsets of the data.

Based on these assumptions, we can approximate the summation in Eq. 1 with the maximum term:

$$p(y|x) = \sum_z p(z|x)p(y|z, x) \geq \max_z p(z|x)p(y|z, x) = \frac{1}{K} \max_z p(y|z, x) \tag{2}$$

Taking the negative logarithm, and omitting the constant $\log K$ term, we get our loss function:

$$\mathcal{L}_{\text{Hard-MoE}}(\theta) = \mathbb{E}_{(x,y)\sim\text{data}} \left[ \min_z - \log p(y|z, x; \theta) \right] \tag{3}$$

To train with this objective, for each mini-batch we enumerate all $K$ possible values of $z$ (typically $K = 10$) and compute their losses: $-\log p(y|z, x; \theta)$. Next, for each $(x, y)$ pair we select the value of $z$ that yields the minimum loss, and use that value to back-propagate, i.e., only one expert receives gradients per sentence pair. Since we make hard selections of the latent variable, we call this method Hard-Mixture of Experts (Hard-MoE). This training algorithm can also be seen as a stochastic gradient descent version of the Hard-Expectation Maximization algorithm.

### 3.2 PRACTICAL CONSIDERATIONS

So far our discussion has been very general. Notice however that our motivation for training with a mixture of experts is different from existing MoE literature that typically aims to increase model capacity (Collobert et al., 2003; Shazeer et al., 2017b; Gross et al., 2017). Instead, we aim to better model multi-modal output distributions—a difference that has significant practical implications.

First, even on large scale MT benchmarks, models are prone to overfit and require regularization (e.g., dropout, label smoothing, etc.); the standard modeling choice of using independently parameterized (decoder) experts is likely to exacerbate this overfitting. Given our goal is not to increase model capacity, we choose instead to share almost all parameters among the experts, except for a single set of weights that we use to embed the different values of the latent variable. Accordingly, we require only a negligible number of additional parameters over a baseline model. More specifically, we found good results in pilot experiments by simply replacing the first dummy beginning-of-sentence or ⟨bos⟩ token of the decoder with the embedded representation of our latent variable.[2] This modeling choice further mitigates the latent variable collapse issue often reported in the MoE literature (Eigen et al., 2014; Shazeer et al., 2017b), in which low quality experts are neglected and eventually "die" during training. Instead, by sharing parameters, even unpopular experts receive some gradients throughout training.

Another potential problem is with dropout. Dropout noise can mask the effect of the change of the latent variable, making assignments inconsistent and eventually leading to a degenerate model

---

[2]We also tried other parameterizations, including adding or concatenating the latent variable embedding with the input word embeddings at each time step, and injecting it into each decoder layer. We found that they have similar results, therefore we adopt the simplest replacing ⟨bos⟩ strategy.

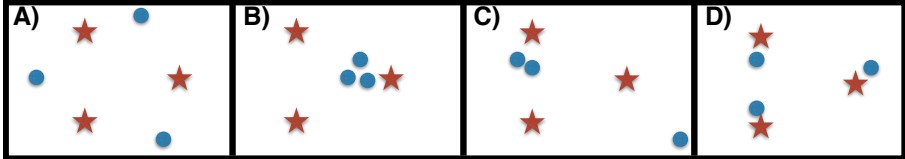

Figure 1: Toy illustration of how the metrics behave in various scenarios. Red stars represent human references, while blue dots represent system hypotheses. **A:** when hypotheses spread uniformly in the space, the average oracle BLEU is low, the coverage is high and the pairwise BLEU is low. **B:** when hypotheses cluster together closely and in vicinity of a reference (a typical case of beam search), the average oracle BLEU is high, the coverage is low and the pairwise BLEU is high. **C:** when there is a poor hypothesis in the set, the average oracle BLEU is low, the coverage is medium and the pairwise BLEU is medium. **D:** when hypotheses match references, the average oracle BLEU is high, the coverage is high and the pairwise BLEU matches human pairwise BLEU.

that ignores the latent variable. Therefore, in practice we decompose the minimization over $z$ in Eq. 3 into two separate steps. First we set the model in *evaluation* mode, turn off dropout, disable gradient computation and perform $K$ forward passes to identify the optimal assignment of $z$.[3] Once the optimal latent values are selected, we switch to *train* mode, turn on dropout, and do another forward-backward pass to update the parameters. This is crucial; in Sec. 5.4 we show that if we instead leave dropout on during minimization, then the latent variable is ultimately ignored.

The whole process is formally described in Algorithm 1. All the decoding strategies for $p(y|x;\theta)$ mentioned in Sec. 2 can be applied to the latent variable model $p(y|z,x;\theta)$ as well. We adopt the most straightforward one: generating $K$ hypotheses by first enumerating $z$ and then greedily decoding $\hat{y}_t = \arg\max_y p(y|\hat{y}_{1:t-1}, z, x; \theta)$. This tests whether our latent variable captures different modes in $p(y|x)$, and we can generate hypotheses efficiently in parallel.

## 4  METHODOLOGY TO ASSESS QUALITY AND DIVERSITY

In this section we describe the metrics we use to quantitatively assess the quality and diversity of a set of translation hypotheses. We use BLEU (Papineni et al., 2002) as a basis for measuring the similarity of an ordered sentence pair $(u, v)$, where $u$ is the reference and $v$ is the hypothesis.

Suppose $\{y^1, \cdots, y^N\}$ are $N$ reference translations of a source sentence $x$, and $\{\hat{y}^1, \cdots, \hat{y}^K\}$ are $K$ hypotheses. We compute the following metrics (Ott et al., 2018a):

- **Average oracle BLEU:** We pair each hypothesis $\hat{y}^k$ with its best matching human reference that gives the highest sentence-level[4] BLEU: $a_k = \arg\max_j \text{BLEU}(y^j, \hat{y}^k)$. Then we compute the corpus-level BLEU of $\{(y^{a_k}, \hat{y}^k)\}_{x \in \text{data}, k \in [K]}$. This measures the *overall quality* of a hypothesis set. If this metric scores low, it implies that some generated hypotheses do not match any reference and the generation quality is poor,

- **Reference coverage:** We pair each hypothesis to its best matching reference (breaking ties randomly), count how many distinct references are matched to at least one hypothesis, and average this count over all sentences in the test set. A low coverage number indicates that all hypotheses are close to a few references. Instead, we would like a diverse set that covers most of the references.

- **Pairwise BLEU:** We compare all hypotheses against each other, and compute the corpus-level BLEU of $\{(\hat{y}^j, \hat{y}^k)\}_{x \in \text{data}, j \in [K], k \in [K] \setminus j}$. This metric measures the similarity among the hypotheses. The more diverse the hypothesis set, the lower the pairwise BLEU. Ideally, we would like a model with pairwise BLEU matching human pairwise BLEU.

---

[3]By putting the model in evaluation mode during minimization we also speed up training and reduce memory consumption, since the $K$ forward passes have no gradient computation or storage. With $K = 10$, we train our model with the same batch size in approximately twice the time as the baseline NMT model.

[4]Sentence-level BLEU is computed with smoothed $n$-gram counts (+1) for $n > 1$ (Ott et al., 2018a).

In addition, we compute the average oracle BLEU and pairwise BLEU of human translations for comparison. The average oracle BLEU is computed in a leave-one-out manner.[5] Namely, each $y^n$ is paired with the best of the rest: $b_n = \arg\max_{j \neq n} \text{BLEU}(y^j, y^n)$, and we compute the corpus-level BLEU of $\{(y^{b_n}, y^n)\}_{x \in \text{data}, n \in [N]}$. A toy illustration of how these metrics behave in different situations is shown in Fig. 1.

## 5 EXPERIMENTS

### 5.1 DATASETS

We test our latent variable models and baselines on three WMT datasets that have multiple human references (Ott et al., 2018a; Hassan et al., 2018). We also use the small IWSLT'14 De-En dataset (Cettolo et al., 2014) for pilot experiments. While IWSLT'14 does not have multiple references, pairwise BLEU lets us test whether the outputs are diverse or not.

**WMT'17 English-German (En-De):** We train on all available bitext excluding the ParaCrawl corpus and filter sentence pairs that have source or target longer than 80 words, resulting in 4.5M sentence pairs. We use the Moses tokenizer (Koehn et al., 2007) and learn a joint source and target Byte-Pair-Encoding (Sennrich et al., 2016) with 32K types. We develop on newstest2013 and test on a 500 sentence subset of newstest2014 that has 10 references (Ott et al., 2018a).

**WMT'14 English-French (En-Fr):** We borrow the setup of Gehring et al. (2017) with 36M training sentence pairs and 40K joint BPE vocabulary. We validate on newstest2012+2013, and test on a 500 sentence subset of newstest2014 that has 10 references (Ott et al., 2018a).

**WMT'17 Chinese-English (Zh-En):** We pre-process the training data following Hassan et al. (2018) which results in 20M sentence pairs, 48K and 32K source and target BPE vocabularies respectively. We develop on devtest2017 and report results on newstest2017 with 3 references.

**IWSLT'14 German-English (De-En):** We follow the setup of Edunov et al. (2018) that has 160K training sentence pairs and 14K joint BPE vocabulary. The test set here has only one reference.

### 5.2 EXPERIMENTAL DETAILS

We build on the Transformer (Vaswani et al., 2017) implementation in the Fairseq toolkit.[6] The encoder and decoder have 6 blocks. The number of attention heads, embedding dimension and inner-layer dimension are 16, 1024, 4096 on the WMT datasets and 4, 512, 1024 on the IWSLT dataset. For variational NMT (Zhang et al., 2016) we use a Gaussian latent variable with $K$ dimensions matching the number of categories of a multinomial latent variable (typically $K = 10$). For baselines involving extra hyper-parameters such as diverse beam search (Vijayakumar et al., 2016) and top-$k$ sampling, we perform grid search over these hyper-parameters and report the best result. Models are optimized with Adam (Kingma & Ba, 2015) using $\beta_1 = 0.9$, $\beta_2 = 0.98$, and $\epsilon = 1e - 8$ and we use the same learning rate schedule as Ott et al. (2018b). We run experiments on DGX-1 machines with 8 Nvidia V100 GPUs and machines are interconnected by Infiniband. We train models with 16-bit floating point operations and mini-batches of approximately 400K tokens for the WMT datasets and 25K tokens for IWSLT, following Ott et al. (2018b).

### 5.3 RESULTS

Table 1 summarizes our main results on three multi-reference large scale datasets, comparing our Hard-MoE model with baselines that do not use a latent variable. To better visualize these numbers, we plot pairwise BLEU in reverse order vs. average oracle BLEU in Fig. 2. The ideal algorithm generates diverse and high-quality translations, i.e., low pairwise BLEU (diverse) as well as high average oracle BLEU (high quality). First, we observe that beam search's average oracle BLEU is fairly close to human's average oracle BLEU, implying a remarkably high generation quality. However, it severely lacks diversity, as shown by the high pairwise BLEU near 80. This suggests a

---

[5]Human score is slightly underestimated in this manner, as each reference is evaluated against other $N - 1$ references, while each system hypothesis is evaluated against $N$ references.

[6]https://github.com/pytorch/fairseq

|  | pairwise BLEU | | | avg oracle BLEU | | | #refs covered | | |
|---|---|---|---|---|---|---|---|---|---|
|  | en-de | en-fr | zh-en | en-de | en-fr | zh-en | en-de | en-fr | zh-en |
| Sampling | 24.1 | 32.0 | 48.2 | 30.3 | 37.6 | 17.7 | 4.6 | 4.3 | 1.5 |
| Beam | 73.0 | 77.1 | 83.4 | 56.5 | 66.6 | 30.7 | 3.1 | 3.2 | 1.3 |
| Diverse beam | 53.7 | 64.9 | 66.5 | 48.2 | 60.2 | 28.6 | 3.7 | 3.5 | 1.4 |
| Hard-MoE | 50.2 | 64.0 | 51.6 | 51.0 | 61.9 | 28.9 | 4.0 | 3.7 | 1.6 |
| Human | 35.5 | 46.5 | 25.3 | 56.7 | 72.0 | 33.6 | - | - | - |

Table 1: Results on the WMT datasets. The En-De, En-Fr, Zh-En datasets have 10, 10, 3 human references respectively, and we generate the same number of hypotheses as the number of references.

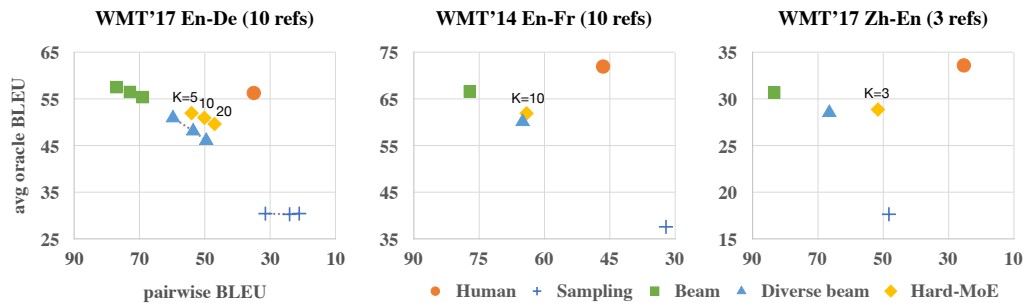

Figure 2: Average oracle BLEU (quality) versus pairwise BLEU (diversity) for various methods on three benchmark datasets with multiple human references. Our approach, Hard-MoE gives the best trade-off.

scenario similar to Fig. 1-B. In contrast, sampling is very diverse, typically even more so than human references on En-De and En-Fr, but has a very poor average oracle performance, as illustrated in Fig. 1-A. Diverse beam search increases the diversity of beam search to some extent, but pays a cost in translation quality. Overall, Hard-MoE achieves the best trade-off between quality and diversity, measured in terms of both pairwise BLEU and reference coverage.

We also explore the impact of varying $K$ on the WMT En-De dataset (Fig. 2, *left*). In general, when more hypotheses are generated (e.g., $K = 20$), they become more diverse but of worse quality. The complete results from these experiments and the sampling top-$k$ baseline are provided in Table 5 in the appendix. Finally, we observe that despite the improvement offered by Hard-MoE, there is still a large gap between the diversity of human references and system generations.

## 5.4 DETAILED COMPARISONS AND ABLATION STUDY

In this section we compare the Hard-MoE model to other latent variable models, including variational NMT and Soft-MoE, on the IWSLT'14 De-En dataset with $K = 2$. As Table 2 shows, none of these alternatives is able to make effective use of the latent variable. Variational NMT ignores the latent variable, demonstrated by its extremely high pairwise BLEU, which is a well-known failure of VAEs on text (Bowman et al., 2016).[7] Next, we tried several variants of Soft-MoE. If expert decoders have separate parameters (Lee et al., 2016), the model collapses and uses only one expert. This is due to the "rich gets richer" effect, whereby only those experts that already explain the targets well receive gradients, causing the other experts to starve during training and eventually die. While there are some "re-balancing" heuristics to alleviate these issues (Eigen et al., 2014; Shazeer et al., 2017a), training is difficult and collapses are often still experienced.

If we instead share the encoder and decoder parameters and disable dropout, Soft-MoE works but severely overfits, resulting in low translation quality (low BLEU). Finally, if we enable dropout during the latent variable selection process, then the model again gets 99 pairwise BLEU. We surmise that this is due to dropout noise masking the effect of selection of different latent values. This pre-

---

[7]We also tried annealing the KL term, but even then the latent variable is hardly used.

|  | pairwise BLEU | BLEU (1st hyp / 2nd hyp) | Failure type |
|---|---|---|---|
| Beam search | 84.8 | 33.5 / 32.4 | - |
| Variational NMT (Zhang et al., 2016) | 99.2 | 32.5 / 32.5 | Latent is ignored |
| Soft-MoE (dropout, separate decoders) | - | 32.7 / 0.0 | Collapse to single expert |
| Soft-MoE (no dropout, shared decoder) | 57.2 | 21.1 / 20.6 | Low translation quality |
| Soft-MoE (dropout, shared decoder) | 99.0 | 32.6 / 32.5 | Latent is ignored |
| Hard-MoE (dropout during minimization) | 99.1 | 32.8 / 32.7 | Latent is ignored |
| Hard-MoE | 72.5 | 32.1 / 32.7 | Success |

Table 2: Results of various latent variable models on the IWSLT'14 De-En dataset. For Variational NMT the hypotheses are generated by first sampling $z$ followed by greedy decoding. Since we only have one reference here, we report pairwise BLEU and BLEU of each hypothesis ($K = 2$) with respect to the reference.

| | | |
|---|---|---|
| Source | 参与投票的成员中，58% 反对该合同交易。 | 自11 月份开始，俄罗斯民意也有所扭转。 |
| Refs | It was rejected by 58 % of its members who voted in the ballot . | Russian public opinion has also turned since November . |
| | Of the members who voted , 58 % opposed the contract transaction . | Russian public opinion has started to change since November . |
| | Of the members who participated in the vote , 58 % opposed the contract . | The polls in Russian show a twist turn since the beginning of November . |
| | | |
| Beam | Fifty-eight per cent of those voting opposed the contract deal . | Since November , public opinion in Russia has also shifted . |
| | Fifty-eight per cent of the voting members opposed the contract deal . | Since November , public opinion in Russia has also reversed . |
| | Fifty-eight per cent of the voting members opposed the contract transaction . | Since November , Russian public opinion has also shifted . |
| | | |
| Diverse beam | Of the members voting , 58 per cent opposed the contract deal . | Since November , public opinion in Russia has also shifted . |
| | Of the members voting , 58 per cent opposed the contract deal . | Since November , the mood in Russia has also changed . |
| | Of the members voting , 58 per cent opposed the transaction . | Since November , public opinion in Russia has also been reversed . |
| | | |
| Hard-MoE | Fifty-eight per cent of the members who voted opposed the contract deal . | Since November , opinion in Russia has also reversed . |
| | Of the members who voted , 58 % opposed the deal . | Since November , the mood in Russia has also reversed . |
| | Fifty-eight per cent of the voting members opposed the contract deal . | Opinion in Russia has also shifted since November . |

Table 3: Examples of generations by different methods on the WMT'17 Zh-En dataset. Our Hard-MoE model shows considerable diversity compared to beam search and diverse beam search.

vents the gating from training well and the latent variable embeddings from specializing, causing the model to essentially ignore the latent variable. We confirm this empirically in the appendix.

## 5.5 QUALITATIVE ANALYSIS

In this section we perform a qualitative analysis with the WMT'17 Zh-En dataset, which contains three reference human translations for each source sentence. In Table 3 we show two source sentences, the corresponding reference translations, and generated hypotheses for several approaches. We observe that beam search tends to produce generations which only differ in the last few words. Diverse beam search improves the diversity over beam search, but is not as diverse as Hard-MoE and may produce duplicate hypotheses (e.g., if the diversity penalty is not sufficiently high). Hard-MoE shows significant diversity in wording, word order, clause structure, etc.

To investigate whether the latent variable in Hard-MoE learns different translation styles, we examine the hypotheses that each latent variable assignment generates. For each value of $z$, we compute word frequencies of the corresponding generations and look for words whose frequency is significantly different as we change the value of the latent variable. We first discover that for words like was, were and had, $z = 1$'s frequency is more than three times higher than $z = 3$'s; conversely, for has and says, $z = 3$'s frequency is more than twice higher than $z = 1$'s. Since Chinese does not have tense unless a time phrase is explicitly stated, we speculate that when translating into English, the first latent value tends to translate with past tense whereas the third latent value tends to translate with present tense. Indeed we find that this is a consistent behavior, as seen from the first four examples in Table 4. Similarly, we find that different latent values exhibit different preferences for using this or that (see the fourth and fifth examples in Table 4), % or per cent (see the last example in Table 4 and the first example in Table 3), and so on.

| Source | 这是一次规模巨大的作业，同时也是一次非常精密的作业。 | 他从不愿意与家人争吵。 |
|---|---|---|
| Reference | This was a massive and , at the same time , very delicate operation . | He never wanted to be in any kind of altercation . |
| Hard- | It was a huge job , and a very delicate one as well . | He never liked to quarrel with his family . |
| MoE | It 's a very large job , and it 's a very delicate one , too . | He never wants to quarrel with his family |
|  | This is a huge job , but also a very delicate one . | He never likes to argue with his family |
| Source | 不断的恐怖袭击显然已对他造成很大打击。 | 我不想说这是我的最后一场比赛。 |
| Reference | Repeat terror attacks on Turkey have clearly shaken him too . | I didn 't want to say this was my last race . |
| Hard- | The continuing terrorist attacks had apparently hit him hard . | I didn 't want to say it was my last game . |
| MoE | He is clearly already being hit hard by the continuing terrorist attacks . | I don 't want to say it 's my last game , he said . |
|  | Repeated terrorist attacks have apparently hit him hard . | I don 't want to say this is my last game . |
| Source | 由此判断，这无疑是一场持续战。 | 两人2015 年缴纳了20.3% 的联邦税。 |
| Reference | It appears that this was definitely an ongoing battle . | They paid a federal effective tax rate of 20.3 percent in 2015 . |
| Hard- | Judging by that , it is undoubtedly a continuing battle . | Both paid a federal tax of 20.3 per cent in 2015 . |
| MoE | It is a battle that is no doubt ongoing . | They paid a federal tax of 20.3 % in 2015 . |
|  | Judging by this , this is undoubtedly a continuing battle . | Both paid 20.3 % of federal taxes in 2015 . |

Table 4: Examples of generations by Hard-MoE on the WMT'17 Zh-En dataset. Different latent values learn to specialize for different translation styles consistently across examples, such as past tense vs. present tense, `this` vs. `that`, and `per cent` vs. `%`.

# 6 RELATED WORK

The training objective of our Hard-MoE approach is reminiscent of the Multiple Choice Learning (MCL) setting for ensemble learning (Guzman-Rivera et al., 2012), where the oracle loss is minimized for an ensemble of learners. Lee et al. (2016) develop this idea and propose a SGD based algorithm to train an ensemble of deep networks for object recognition tasks. In another work, Lee et al. (2015) introduce a tree structure parameter sharing of a CNN ensemble. Our method shares the same motivation and underlying algorithm, but the architecture and the task are very different. Moreover, our goal is to produce a diverse and high-quality set of translations, which we evaluate using multiple reference human translations and several quantitative metrics, as opposed to just evaluating the best matching hypothesis.

There have been several prior studies investigating uncertainty in machine translation. Dreyer & Marcu (2012) and Galley et al. (2015) introduced new metrics to address uncertainty at evaluation time. Ott et al. (2018a) inspected the sources of uncertainty and proposed tools to check fitting between the model and the data distributions. They also observed that modern conditional auto-regressive NMT models do capture uncertainty, but only to a certain extent, and they tend to over-smooth probability mass over the hypothesis space.

In addition to the variational NMT model (Zhang et al., 2016), there are two extended models introducing latent variables in NMT. Schulz et al. (2018) considered a sequence of latent Gaussian variables to represent each target word in an auto-regressive fashion, as opposed to one latent variable representing the whole target sentence. Kaiser et al. (2018) proposed a similar model, except that they used groups of discrete multinomial latent variables. In their qualitative analysis, Kaiser et al. (2018) showed that the latent codes do affect the output predictions in interesting ways, but their focus was on speeding up regular decoding rather than producing a diverse set of hypotheses. Concurrent to our work, He et al. (2018) consider the same problem and a similar approach to train a Soft-MoE model. They propose to sample a latent value from the posterior distribution to estimate the gradients efficiently, as opposed to we select a latent value via minimization as done in this work. Moreover, they do not consider evaluation datasets with multiple references and use weaker baseline models, which makes their empirical analysis less conclusive.

Besides machine translation, there are several works introducing latent variables for dialogue generation (Serban et al., 2017; Cao & Clark, 2017; Wen et al., 2017) and image captioning (Wang et al., 2017; Dai et al., 2017). Our method departs from these VAE or GAN-based approaches and importantly, is much simpler. It could also be applied to other text generation tasks as well.

# 7 CONCLUSION

We propose a hard mixture of experts model to generate a diverse and high-quality set of hypotheses for machine translation. This model is very simple and it can be used as a strong baseline for other

approaches involving latent variables. Although mixture of experts models are powerful, they are very brittle at training and are prone to several failure modes, such as using only a small subset of experts or learning undifferentiated experts, particularly when working with large, heavily regularized models as in NMT. Our Hard-MoE approach makes several simplifying assumptions on top of the baseline MoE algorithm, which effectively mitigates these kinds of failures.

We report results from extensive experiments on three large-scale datasets with multiple references, and demonstrate that Hard-MoE achieves the best trade-off between quality and diversity. Still there is a big gap between our model performance and human performance. In the future, we would like to build upon this work to investigate models with richer and more structured latent representations.

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

# A    DETAILED RESULTS

| | pairwise BLEU | | | avg oracle BLEU | | | #refs covered | | |
|---|---|---|---|---|---|---|---|---|---|
| | $K = 5$ | 10 | 20 | 5 | 10 | 20 | 5 | 10 | 20 |
| Sampling | 31.6 | 24.1 | 21.2 | 30.5 | 30.3 | 30.5 | 3.1 | 4.6 | 6.2 |
| Filtered sampling (top-2) | 49.3 | 47.8 | 46.7 | 47.0 | 47.3 | 47.5 | 2.7 | 3.7 | 4.7 |
| Beam | 77.1 | 73.0 | 69.1 | 57.6 | 56.5 | 55.4 | 2.3 | 3.1 | 4.0 |
| Diverse beam | 59.8 | 53.7 | 49.7 | 51.0 | 48.2 | 46.1 | 2.5 | 3.7 | 4.8 |
| Hard-MoE | 54.2 | 50.2 | 47.1 | 52.0 | 51.0 | 49.7 | 2.8 | 4.0 | 5.3 |

Table 5: Results on the WMT'17 En-De dataset with various numbers of generations ($K$). We compare: multinomial sampling (`Sampling`); sampling restricted to the top-$k$ candidates at each step (`Filtered sampling (top-2)`); beam search with varying beam widths (`Beam`); diverse beam search (Vijayakumar et al., 2016) with varying number of outputs (`Diverse beam`; note that the number of groups $G$ and diversity strength are tuned separately for each value of $K$); and our hard mixture of experts approach with $K$ latent categories (`Hard-MoE`).

In Table 5 we compare different approaches for generating diverse translations on the WMT'17 En-De dataset. We additionally compare each approach as we vary the quantity of desired translations ($K$). We observe that sampling produces diverse but low quality outputs. We can improve translation quality by restricting sampling to the top-$k$ candidates at each output position ($k = 2$ performed best), but translation quality is still worse than hard-MoE. Beam search produces the highest quality outputs, but with low diversity. Diverse beam search provides a reasonable balance between diversity and translation quality, but the hard-MoE approach produces even more diverse and better quality translations. Finally, besides unrestricted sampling, hard-MoE covers the largest number of references of any of the approaches evaluated.

# B    EFFECTS OF DROPOUT DURING MINIMIZATION

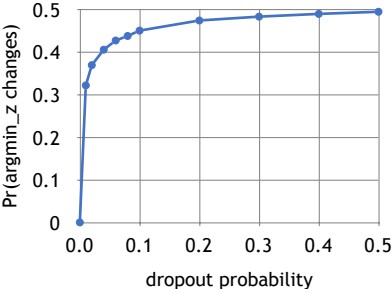

Figure 3: The effect of dropout on the minimization process at the beginning of training. Experiments are performed on the IWSLT'14 De-En dataset with 2 latent categories ($K = 2$).

In Section 5.4 we observed that it is crucial to turn off dropout when performing the minimization over $z$. We speculate that the reason is that dropout noise dilutes the impact of the latent variable, causing the minimization process to select among the latent values at random. Then it prevents different latent values from specializing and ultimately causes the model to ignore them.

To test this hypothesis, we show in Fig. 3 the effect of dropout noise on the minimization process at the beginning of training (i.e., with a randomly initialized model). On the y-axis we plot the probability that the optimal value of $z$ changes after applying dropout with different probabilities. We see that as we increase the dropout probability, the optimal value of $z$ is quickly corrupted—even with a dropout probability of 0.1 we observe a 45% chance that the optimal assignment of $z$ changes.

