# OpenReview forum: "Diverse Machine Translation with a Single Multinomial Latent Variable"
_ICLR.cc/2019/Conference_

### Official Review · AnonReviewer2 · 2018-11-03
**Interesting contribution**

**Rating:** 7
**Confidence:** 4

**Review:**

This paper proposes a sequence to sequence model augmented with a multinomial latent variable. This variable can be used to generate multiple candidate translations during decoding. This approach is simpler than previous work using continuous latent variables or modifying beam search to encourage diversity, obtaining more diverse translations with a smaller drop in translation accuracy.

Strengths:
- Simple model that succeeds in achieving its goal of generating diverse translations.
- Provides insights into training models with categorical latent variables.
Weaknesses:
- More insight into what the latent variable is learning to represent would strengthen the paper.

While the model is simple, its simplicity has significant strengths: In contrast to more complex latent space, the latent variable assignments can be enumerated explicitly, which enables it to be used to control the generation and compare outputs. The simplicity of the model will force the latent variable towards capturing diversity - modelling uncertainty in how to express the output rather than uncertainty in the content.

One question about the model architecture is just whether it is sufficient to feed the latent variable embedding only once, as it effect might be diluted across long output sequences (as opposed to, say, feeding the latent variable at each time step).

The paper provides some interesting insights, such as the need to do hard EM-style training and turning off dropout when inferring the best latent variable assignment during training, to avoid mode collapse.

What is the effect of initialization? This often has a large impact in EM-style training, and could also lead to mode collapse, though in this case the restricted parameterization might prevent that.

What is the training time and computational resource requirements? Are multiple DGX-1s running in parallel required to train the model?

What is not clear enough from the paper is what kind of structure the latent variables learn to capture. In particular this model is not biassed towards any explicit notion of the kind of diversity one would like to learn. While there is some qualitative analysis, further analysis would strengthen the paper.

Overall this is a very interesting contributions that offer useful insights into designing controllable sequence generation models.

---

> ### Author Response · Authors · 2018-11-15
> **Response to AnonReviewer2**
>
> We thank the reviewer for the feedback and comments.
>
> “whether it is sufficient to feed the latent variable embedding only once”
> In addition to feeding the latent variable embedding once as the beginning-of-sentence (<bos>) token, we also tried other model architectures, including adding or concatenating the latent variable embedding with the input word embeddings at each time step, and injecting it into each decoder layer. We found that they have similar results, therefore we adopt the simplest replacing <bos> strategy. We hypothesize that the latent variable's effect does not dilute across long output sequences because the decoder in Transformer has a self-attention mechanism. When generating each word it can always attend to the latent variable directly without being affected by distance.
>
> “Effect of initialization”
> In our experiments, we randomly initialize all weights following the reference Transformer implementation, and our findings are quite consistent across random seeds. As the reviewer conjectures, the particular choice of parameter sharing between experts is key to avoid the common failure mode of one expert taking over all the others, as their parameters are always updated even when they are not selected for a particular input example.
>
> “What is the training time and computational resource requirements? Are multiple DGX-1s running in parallel required to train the model?”
> Training the Hard-MoE model with 10 states takes about twice as long as training the baseline without latent variable. We train the big WMT En-De model with 10 latents in ~3.5 hours on 128 GPUs. An equivalent model can be trained on a single machine with 8 GPUs in ~1.5 days (due to faster inter-GPU communication), or with 1 GPU in about a week.
>
> “Structure captured by the latent variable?”
> The reviewer is correct that in the proposed approach there is no constraint on what each latent value represents.
> As a result the different translation styles captured by each latent value are often mixed and have no clear structure. From Sec. 5.5 and Table 4 we can see that z=1 captures past tense, “that” and “per cent”, while z=3 captures present tense, “this” and “%”, for instance. We are working towards adding a similar analysis on the En-De dataset, following your suggestion.
> In general, we would like to investigate models with richer and more structured latent representations in future work.

---

### Official Review · AnonReviewer1 · 2018-11-03
**Review for Diverse MT with a Single Multinomial Latent Variable**

**Rating:** 5
**Confidence:** 4

**Review:**

This paper studies the diverse text generation problem, specifically on machine translation problem. The authors use a simple method, which just using a single multinomial latent variable compared with previous approaches that using multi latent variables. They named the approach: Hard-MoE. They use parallel greedy decoding to generate the diverse translations and the experiments on three WMT datasets show the approach make a trade-off between diversity and quality.
In general, I think generating the diverse translations for machine translation problem may not so important and piratically in actual scenarios. In fact, how to generate fluent and correct translations is more important.

For the details, there are some problems. 1) The only modification for this work is to make the soft probability of p(z|x) to be 1/K. The others are several experimental studies. To be an formal ICLR paper, this may not be interesting enough to draw my attention. 2) In case of the results, though the authors claimed they achieved better trade-off between diversity and quality, in my opinion, the beam original beam search is good enough from the results in Table 1. 3) In table 2, what means k=0 for the BLEU score? 4) I want to indicate that the purpose of VAE approach related to this work is to increase the model performance w.r.t. the BLEU score instead of the diversity, same as the original MoE method. 5) There are some related works to this work, but their methods are also very effective in terms of the BLEU score, e.g., the author can check this one in EMNLP this year: “Sequence to Sequence Mixture Model for Diverse Machine Translation”. Authors may need a more discussion between those works and this work.

---

> ### Author Response · Authors · 2018-11-15
> **Response to AnonReviewer1**
>
> “generating the diverse translations for machine translation problem may not so important and piratically in actual scenarios”
> MT systems are approaching human level performance on several language pairs (see recent results of WMT competitions, for instance). Improving translation quality remains a big challenge on low resource languages, but not as much on high resource languages such as those considered in this work. Instead, providing the user with a diverse set of translations that capture different translation styles (e.g., formal/informal, literal/not literal) and information asymmetry between different languages (e.g., when translating from a language without tense to a language that requires tense specification) may become the next important feature of MT systems.
> More generally, the study of how to model uncertainty is key to the advance of AI systems in non-deterministic settings and for prediction tasks that are inherently multi-modal. The MT task serves as a good test bed application for this. See the first two paragraphs of the introduction.
>
> “1) The only modification for this work is to make the soft probability of p(z|x) to be 1/K.”
> This statement is not correct. As we explain in Sec. 3.1, in addition to the change in prior we also introduce a new training procedure that includes inference of the latent variable and selectively backpropagating through only the optimal latent variable assignment.
>
> “2) ... the beam original beam search is good enough from the results in Table 1”
> This statement is not correct. Comparing the “pairwise BLEU” (lower numbers indicate more diversity) and “#refs covered” columns in Table 1, we show that beam search is about 20 (pairwise) BLEU points less diverse than our approach! The qualitative results in Table 3 further illustrate the significant improvement in diversity of our approach compared to beam search.
>
> “3) In table 2, what means k=0 for the BLEU score?”
> In this experiment, we use different models to generate 2 hypotheses and compute the BLEU score of the first generated hypothesis (k=0) and the second generated hypothesis (k=1) w.r.t. the reference respectively. We have changed the notation to make this clearer, see new table 2.
>
> “4)  I want to indicate that the purpose of VAE approach related to this work is to increase the model performance...”
> A general discussion about the purpose of latent variable models is beyond the scope of this rebuttal.
> In short, latent variables are one major and principled approach to describe uncertainty of distributions, and VAE is a density estimation framework equipped with latent variables. The conditional distribution we aim to model has uncertainty (as there are several plausible ways to translate a source sentence). Such uncertainty is partly captured  by the output distribution of the decoder and partly modeled by the stochastic latent variables.
> The potential advantage of VAEs may stem from the implicit regularization induced by the use of latent variables at training time (which is what the reviewer seems to be referring to) or from the better modeling of the underlying uncertainty (which is what we are studying in this paper).
>
> 5) Relation to “Sequence to Sequence Mixture Model for Diverse Machine Translation”
> This paper is concurrent to ours and was not available at submission time. They also consider the same problem and propose a similar approach. The major differences are:
> a) we select only one expert while they use all of them. This means that their model is much more expensive in terms of memory and computation at training time; in fact for larger number of latent states they also propose to select one latent value but they do so by sampling as opposed to via minimization.
> b) the parameterization is different.
> c) we study the collapses of latent variable models and provide insights on how to prevent these failures.
> d) their evaluation is limited to the small IWSLT dataset and to small baseline models, while we use the much bigger WMT datasets, state-of-the-art baselines and we leverage multiple references for each source sentence in our evaluation. Our paper proposes metrics and reports a more in depth analysis of diversity both quantitatively and qualitatively.
> We have added a reference to this paper in the revised version. Thank you for the suggestion.

---

### Official Review · AnonReviewer3 · 2018-11-04
**Interesting but somewhat incremental approach; related work a bit weak**

**Rating:** 6
**Confidence:** 4

**Review:**

The authors aim to increase diversity in machine translation using a multinomial latent variable that captures uncertainty in the target sentence. Modeling uncertainty with latent variables is of course relatively common in ML, and this work has similarities with latent variables models for MT [Zhang et al., 2016] and for other generation tasks such as dialogue [Serban et al., 2017; etc.]. The key difference is that the authors here use a Mixture of Expert (MoE) approach while most relevant prior works use variational approaches. Experiments show improvements in diversity over variational NMT [Zhang et al., 2016] and decoding-time approaches (e.g., diversity constraints [Vijayakumar et al., 2016]).

Overall, the proposed approach (hard-MoE) is well motivated and the experimental results are relatively promising. I think the authors did a good job analyzing and justifying their approach against the soft version of their model (i.e., soft-MoE causes experts to “die” during training) and variational alternatives (i.e., variational approaches often have failure modes where the latent variable is effectively ignored.)

However, I find related work a bit weak because the problem of producing diverse output has been a much bigger focus in tasks other than MT, such as dialogue and image captioning. The paper glosses over related approaches on these tasks, but the need to model uncertainty for these other tasks is much bigger since source and target are usually not semantically equivalent. So it would have been nice to see argumentative (or even empirical) comparisons with popular models such as VHRED for dialogue [Serban et al., 2017], as many of these models are not intrinsic to either MT or dialogue (the only aspect specific to dialogue in VHRED is context, but it can be set to empty and thus VHRED could have been used as a baseline in the paper.) It would be interesting to compare the work against Serban et al. [2017]’s justification for using a latent variable, which is quite different (see their bit on “shallow generation”, and the idea that their latent variable encapsulates “the high-level semantic content of of the output”).

One technical caveat is that there appears to be some inconsistency in the comparison between human and systems in Table 1. If N is the number of references, then systems are evaluated on N references while the human “system” on only N-1 because of leave-one-out. While this difference might have less of an impact on “average oracle BLEU” than standard BLEU, having one less reference might still penalize the human “system”, and this might partially explain why “beam search’s average oracle BLEU is fairly close to human’s average oracle BLEU”. The right thing to do would be to evaluate both human and all systems in a leave-one-out approach (i.e., let references [r1 … rN] and systems [s1 … sM], then evaluate each element of [s1 … sM r1] on references [r2 … rN], etc.). In that manner, all the “systems” including human are consistently evaluated on *exactly* the same references.

Minor comments:

 “By putting the model in evaluation mode during minimization we also speed up training and reduce memory consumption, since the K forward passes have no gradient computation or storage.” In other words, does this mean the algorithm is easy to *parallelize* because sharing parameters is often what kills the effectiveness of parallelized SGD and variants? If so, “parallelizing” is key word to mention here otherwise I don’t see how we can speed that up by increasing K.

Figure 2: performance drops with K approaching 20. What happens with K=50 or 100 or more? This is a bit of a concern because (1) larger K could require a massive amount parallelization and (2) competing approaches such as VHRED can handle latent variables with higher capacities.

Practical considerations subsection is too vague: parameter sharing is not formally/mathematically explained and the work could be hard to reproduce exactly (as there are often different ways to share parameters).

Why no “#ref covered” for human in Table 1, and why no comparison with Variational NMT? Zhang et al [2016] is the most talked about competing model, so it should probably be evaluated on both settings.

Missed reference: Mutual Information and Diverse Decoding Improve Neural Machine Translation.
Jiwei Li, Dan Jurafsky. https://arxiv.org/abs/1601.00372

---

> ### Author Response · Authors · 2018-11-15
> **Response to AnonReviewer3**
>
> We thank the reviewer for the feedback and comments. We address each of them in turn:
>
> The VHRED model for dialogue (Serban et al., 2017) has a Gaussian latent variable for each utterance given the context and is trained with the VAE objective. Applying this model to machine translation is equivalent to having a latent variable for the target sentence given the source--- which is exactly the variational machine translation baseline (Zhang et al., 2016) we compare to (see Table 2). Our method is not limited to MT and can be applied to other text generation tasks such as dialogue and image captioning. In this work we choose MT because this is an application where the importance of modeling uncertainty has been underestimated so far. People often think that source and target sentences should be semantically equivalent, but neglect the different translation styles (e.g., formal/informal, literal/not literal) and information asymmetry between different languages. For example, Chinese has no tense, while English requires tense specification, and our experiments show that the latent variable captures this phenomenon (Sec. 5.5). More examples include grammatical gender, honorifics, etc. We believe it's important to provide such a variety of translations, which is an unresolved problem in current MT systems. Moreover, compared to dialogue and other applications, MT has a more widely established metric, BLEU, which enables us to do systematic evaluation against a set of reference translations.
>
> The reviewer is right that when computing average oracle BLEU among human references, each reference is evaluated against other N-1 references, while each system hypothesis is evaluated against N references, which could put the score for human at a disadvantage. We have added a note  about this in the updated version.
> We also conducted a leave-one-out “average oracle BLEU” evaluation for system hypotheses, i.e. each hypothesis is evaluated against all N-1 subset of N references, followed by averaging. This is equivalent to pair each hypothesis to its best matching reference N-1 times, and its second best once. On the WMT’17 Zh-En dataset for which we only have 3 human references (and therefore, the two evaluations will differ the most), the results are:
>                                 against all refs (as in the paper)   leave-one-out
>     Sampling           17.7                                                    15.9
>     Beam                 30.7                                                     27.8
>     Diverse beam   28.6                                                    25.8
>     Hard-MoE         28.9                                                     26.1
>     Human                                                                          33.6
> We can see that the relative rankings between the models stay the same, this different evaluation just slightly change the absolute value of this metric. Therefore, all our conclusions hold the same.

---

> ### Author Response · Authors · 2018-11-15
> **Response to AnonReviewer3 (continued)**
>
> “By putting the model in evaluation mode during minimization we also speed up training and reduce memory consumption, since the K forward passes have no gradient computation or storage.”
> Learning a Soft-MoE model with K experts requires K-1 times more work in both the forward and backward pass compared to a baseline (single expert) model. In contrast, our Hard-MoE model requires K times more work in the forward pass (to choose z), but the backwards pass is the same work as the baseline (see Algorithm 1). Moreover, the K forward passes can be very efficiently parallelized since they do not require storing any intermediate values for backpropagation and therefore require less GPU memory (e.g., using the torch.no_grad option in PyTorch).
> With K=10, the training time of our model is roughly twice that of a baseline model on the same hardware. At test time we can generate K hypotheses from each value of the latent variable in parallel via greedy decoding.
>
> “Small/big value of K”
> In this paper we aim at generating order 10 hypotheses, as we think that's a reasonable amount to present to users in practical applications, while still capturing the most significant diversity. This also matches the number of references we have available for evaluation. We leave to future work scaling to a much larger number of states and properly evaluating in that setting. Note that although competing approaches like VHRED or Variational NMT have the potential to model a larger variety of hypotheses by introducing continuous latent variables, they actually fail to use the latent variable and cannot generate different hypotheses from different values of z. It seems that the benefits of those approaches are mostly due to their implicit regularization (due to the addition of noise in the latent space), more than better modeling ability.
>
> “Parameter sharing”
> We share all parameters among the experts, except that each expert has a unique beginning-of-sentence embedding in the decoder (see paragraph 2, Sec. 3.2). We also tried other parameterizations, such as to add or concatenate the latent variable's embedding with the input word embeddings at each time step, and to inject it into each decoder layer, but found similar results. Therefore we adopt the simplest approach described above.
>
> “#ref covered”
> We could divide all human translations into half and half, use one set as reference and the other as hypothesis to compute the coverage number. However this approach doesn't make full use of all human references, and different divisions lead to different numbers.  Besides, “#ref covered” serves the same purpose as “pairwise BLEU”, both of which measure diversity. We therefore consider “pairwise BLEU” as a more direct metric for comparing both the diversity of human references and the diversity of system hypotheses.
> “Variational NMT”
> Variational NMT fails to use the latent variable on the IWSLT dataset. It degenerates to the baseline NMT model and cannot generate different hypotheses (see table 2) unless we use diverse decoding strategies (sampling, beam, diverse beam). Therefore we did not test it on the larger WMT datasets.
>
> “Missing citation”
> The beam search diversification heuristic proposed by Li and Jurafsky (2016) is outperformed by diverse beam search (Vijayakumar et al., 2016), which is a baseline in our paper. It's indeed relevant and we have added a reference to it in our updated version.

---

### Official Review · AnonReviewer4 · 2018-12-03
**Good direction, though problematic assumptions**

**Rating:** 3
**Confidence:** 4

**Review:**

# Summary of model

The paper proposes a mixture model formulation of NMT where the mixing coefficients are uniform and fixed. The authors then proceed to derive a lowerbound on the marginal likelihood

p(y|x) = \sum_z p(z)p(y|x,z) > 1/K \max_z p(y|x,z)

by picking the component z for which the joint likelihood is maximised. With a uniform p(z) this clearly selects the z for which the conditional p(y|x,z) is maximum. I use strictly greater here because p(z) > 0 and p(y|x,z) > 0 for every z.

The loss L(\theta|x,y) for an observation (x,y) is \min_z - \log p(y|z,x; \theta)
whose gradient with respect to NN parameters (theta) is \grad_theta \log p(y|z,x; \theta) for the component z that minimises the negative log-conditional and 0 for every other component, thus while this requires K forward passes (to solve \min_z), it only takes 1 backwards pass.

# Discussion

I appreciate model-based (as opposed to search-based) attempts to improve diversity for generation tasks such as MT. Latent variable modelling aims at a more explicit account of the generative procedure, namely, the joint distribution, which can potentially disentangle and explain different modes of the marginal. Thus from that point of view, this paper points to an exciting direction. That said, in my view, the assumptions behind the proposed approach are not justifiable and some of the claims are simply not appropriate. Below I try to support this view.

A stepping stone of this model is that p(y|x,z) must be "large for only one value of z" (as authors put it), and authors *assume* that will be the case.

While the bound in equation (2) holds, whether or not p(y|z,x) turns out to be "large for only one value of z", it will be a very loose bound unless that happens.

The key point is that one cannot *assume* it to be the case. One could perhaps *promote* it to be the case, but there's no aspect of the model formulation (or objective) that promotes such behaviour.

Backpropagating through whichever component happens to assign the largest likelihood does not guarantee (nor encourages) the other conditionals to *independently* end up going to zero.

Given the level of parameter sharing, I'd even consider the possibility that the exact opposite happens. As authors put it themselves

"Instead, by sharing parameters, even unpopular experts receive some gradients throughout training."

It's true they do, but they are being updated on the basis of the unilateral opinion of the selected component about the likelihood of the data.

Note that the true posterior p(z|x,y) is exactly proportional to the likelihood, as the prior is *uniform and fixed*:
  p(z|x,y) \propto 1/K p(y|x,z) \propto p(y|x,z)
This means that the authors expect the likelihood to do component allocation on its own. That is, the conditionals p(y|x,z=1), ..., p(y|x,z=K) must somehow coordinate themselves in making good use of the latent components. Without any mechanism to promote "competition" (in the parlance of Jacobs et al 1991), I don't see how this can work.

Also, the paper claims to model uncertainty, if I take the posterior to fulfil this claim, then I'm just left with a likelihood (again, due to uniform prior). In any case, a notion of uncertainty here would be conditioned on a point estimate of the network's parameters and should thus be worded carefully.

# Clarifications

1. "we aim to explicitly model uncertainty during training" can you make a case for where that happens in your model?

2. "prevents the gating from training well and the latent variable embeddings from specializing" which gating?

3. "While they showed improvements due to the regularization effect of the Monte Carlo gradient estimate”. I find it strange to talk about the “regularisation effect” of a gradient estimate, perhaps you can be a bit more precise here? Or perhaps you are referring to some specific component of the objective function whose gradient we are estimating via MC and perhaps that component may have some regularisation effect.

4. if you aim to have p(y|x,z) high for a single latent variable at a time, you are implicitly saying that every x has at most (or rather exactly) K translations with non-negligible probability. Is that sensible?

# Pros/Cons

Pros

* simple: the approach presented here requires no significant changes to otherwise standard architectures, it instead concentrates in a change of objective and training algorithm.
* assessment of variability in translation: this paper proposes to use BLEU and a corpus of multiple references in an interesting (potentially novel) way.

Cons

* problematic assumptions: e.g. posterior will turn out sparse without any explicit way to promote such behaviour
* unrealistic claims: e.g. modelling uncertainty
* imprecise use of technical language: some technical terms are not used in their strictly technical sense (e.g. uncertainty, degeneracy), some explanations employ loosely defined jargons (e.g. regularisation effect of the gradient estimate)

---

> ### Author Response · Authors · 2018-12-05
> **To AnonReviewer4 (continued)**
>
> Answers to specific questions:
> 1.  Modeling uncertainty.
> By uncertainty in the output distribution we mean the fact that there are multiple plausible translations of the same sentence (hence, uncertainty in what to predict).
> The baseline model leaves all the uncertainty in the decoder distribution p(y|x) and it is hard to search for multiple modes with this. Instead, we introduce a latent variable to capture some of the uncertainty. If successfully learned, p(y|x,z) is going to have less uncertainty and we can better explore multiple modes of p(y|x) by first sampling/enumerating z and then greedily decoding p(y|x,z). Our objective and training algorithm are designed for this purpose, as discussed above.
>
> 2. which gating
>  It's the gating function p(z|x;theta) of Soft-MoE (Sec. 3).
> We will clarify in the revision.
>
> 3. "While they showed improvements due to the regularization effect of the Monte Carlo gradient estimate”
> The VAE objective is L(theta,phi;x)=E_{q(z|x;phi)}[-log p(x|z;theta)] + D_{KL}(q(z|x;phi)||p(z)), the expected reconstruction error plus a KL regularizer of the posterior. The latter is directly calculated, and the former can be optimized via the reparameterization trick and Monte Carlo gradient estimate (Kingma and Welling, 2013). Applying these to variational NMT (we need to replace x with y and condition on x, since VAE is a model of p(x) and here we're modeling p(y|x)), it collapses and the posterior is always the same as the prior so their KL divergence is 0 (Bowman et al., 2016). Comparing the remaining term E_{z~N(0,I)}[-log p(x|z;theta)] (or -log p(y|x,z;theta)) with traditional loss -log p(x;theta) (or -log p(y|x;theta)) that doesn't involve a latent variable, the MC gradient estimate for the former may bring some regularization effect. So, we are referring to the fact that although latent variables in variational NMT may not be used, they are still useful in the sense that at training time sampling noise may have a regularization effect in the overall model. We briefly mentioned this in the introduction as it's not the main part of this paper.
>
> 4. “if you aim to have p(y|x,z) high for a single latent variable at a time, you are implicitly saying that every x has at most (or rather exactly) K translations with non-negligible probability. Is that sensible? ”
> Going back to K-means, a priori there is no right way to pick K; the same applies in our case. This is admittedly a crude modeling assumption; however, our experiments show that this model does work better than models that are equipped with richer internal representation (like variational NMT) but more prone to degeneracies that ignore the latent variable. We leave to future work exploring more effective modeling choices.

---

> ### Author Response · Authors · 2018-12-05
> **To AnonReviewer4**
>
> We thank the reviewer for the valuable feedback. We realize how the current wording in the paper may have been misleading. Below, we will clarify our approach and try to better support and motivate our modeling choices. We will also update our model description accordingly in the next revision.
>
> First, we provide an alternative interpretation of our lower bound to better illustrate how our objective rewards diversity, and then draw an analogy to K-means to further motivate our approach.
>
> p(y|x) = sum_z p(z|x)p(y|x,z) > 1/K max_z p(y|x,z)
> We agree with the reviewer that this lower bound may be loose. However, the idea is that as we maximize the lower bound, we're maximizing the marginal likelihood p(y|x) minus the gap g=1/K (sum_z p(y|x,z) - max_z p(y|x,z)). We want to learn a model that achieves both large marginal likelihood and small gap, and our formulation of the objective p(y|x)-g=1/K max_z p(y|x,z) enables efficient optimization of such quantity. Our model does NOT assume that p(y|x,z) must be large for only one value of z (we acknowledge that the current wording in the paper is imprecise about this and we will correct it); rather, we are optimizing for this by incorporating g into our loss. g rewards diversification and achieves its minimum when p(y|x,z) is large only for one value of z. Of course, optimizing p(y|x) -g does not guarantee that the individual terms will be optimized as this is not convex, but our loss does reward for g being small.
>
> Another interpretation of our model comes from an analogy to K-mean. The proposed model can be simplified to its core by 1) removing the conditioning on x and 2) assuming that the output space of y is in R^d. In this setting, the soft-mixture of experts reduces to a mixture of Gaussians (MoG). If we replace the prior with a uniform and replace the marginalization with a minimization, as we propose in our model, one recovers K-means. Admittedly, MoGs and K-means are known to possibly fail; after all, the corresponding log-likelihood loss is not convex in this case. For example, it is possible that clusters “die” and it is also possible for poor initialization to yield poor diversification of clusters. Nevertheless, K-means is effective in practice and often produces useful clusterings of the data. Our model is an instance of on-line K-means but in the conditional case and when the output space is the space of word sequences. We find that this model is less prone to collapse and works consistently better than Soft-MoE and VAE on three different language pairs using state-of-the-art architectures and large scale datasets.
>
> Finally, we'd like to emphasize that compared to vanilla K-means in R^d, there are more subtle factors here that are important for the model to be optimized well. For example, we found that implementation details such as the amount of parameter sharing among experts and how dropout is applied can dramatically influence how a model is going to use its latent variable.
>
> Overall, this paper contributes a) a simple (K-means like) model that works on a variety of datasets and outperforms existing approaches by quite some margin, b) a novel evaluation protocol assessing both quality and diversity of generations, and c) code to reproduce all of the empirical findings. We truly believe these are valuable contributions to the community working on text generation, and our model is an important baseline for future work.
>
> We hope through the above discussions the reviewer can appreciate the merits of this work. We will improve clarity and make the suggested changes, and we are always available for further discussions.

---

### Meta-Review · Area_Chair1 · 2018-12-16
**writing needs to be improved / contribution limited**

**Confidence:** 4
**Recommendation:** Reject

**Metareview:**

+ a simple method
+ producing diverse translation is an important problem

- technical contribution is limited / work is incremental
- R1 finds writing not precise and claims not supported,  also discussion of related work is considered weak by R3
- claims of modeling uncertainty are not well supported


There is no consensus among reviewers.  R4 provides detailed arguments why (at the very least) certain aspects of presentations are misleading (e.g., claiming that a uniform prior promotes diversity). R1 is also negative, his main concerns are limited contribution and he also questions the task (from their perspective  producing diverse translation is not a valid task; I would disagree with this).  R2 likes the paper and believes it is interesting, simple to use and the paper should be accepted. R3 is more lukewarm.